# Role of Interdiffusion and Segregation during the Life of Indium Gallium Arsenide Quantum Dots, from Cradle to Grave

**DOI:** 10.3390/nano12213850

**Published:** 2022-10-31

**Authors:** Thomas Walther

**Affiliations:** Department of Electronic & Electrical Engineering, University of Sheffield, Mappin Building, Mappin Street, Sheffield S1 3JD, UK; t.walther@sheffield.ac.uk

**Keywords:** electron microscopy, InGaAs, Stranski-Krastanow, islands, quantum dots

## Abstract

This article summarizes our understanding of the interplay between diffusion and segregation during epitaxial growth of InGaAs and InAs quantum dots. These quantum dots form spontaneously on flat GaAs (001) single-crystalline substrates by the so-called Stranski-Krastanow growth mechanism once a sufficient amount of indium has accumulated on the surface. Initially a perfectly flat wetting layer is formed. This strained layer then starts to roughen as strain increases, leading first to small, long-range surface undulations and then to tiny coherent islands. These continue to grow, accumulating indium both from the underlying wetting layer by lateral indium segregation and from within these islands by vertical segregation, which for InGaAs deposition results in an indium-enriched InGaAs alloy in the centre of the quantum dots. For pure InAs deposition, interdiffusion also results in an InGaAs alloy. Further deposition can lead to the formation of misfit dislocations that nucleate at the edges of the islands and are generally sought to be avoided. Overgrowth by GaAs or InGaAs alloys with low indium content commences preferentially between the islands, avoiding their strained edges, which initially leads to trench formation. Further deposition is necessary to cap these quantum dots effectively and to re-gain an almost flat surface that can then be used for subsequent deposition of multiple layers of quantum dots as needed for many optoelectronic devices.

## 1. Introduction

Quantum dots are small artificial crystals of semiconductors that behave electronically like super-sized atoms and can exist either as nanocrystals in liquids (colloidal form) or embedded in solids (epitaxial form). Of the latter, made by physical or chemical deposition methods, those made up of compounds from groups III and V of the periodic table of the elements (III–V semiconductors) are particularly relevant because many of them have a direct bandgap and they can be alloyed with one another to adjust the bandgap to specific values needed for certain applications in opto-electronics. InGaAs has thus been routinely used to fabricate red and infrared light emitting diodes [1], laser diodes [2,3] and infrared photodetectors [4]. 

For InAs [5] and InGaAs [6] quantum wells or quantum dots on GaAs(001) substrates, the layer widths, quantum dot size, and the spatial distribution of the indium atoms determine both the strain and the optical properties of the quantum structures formed [7,8]. The Stranski-Krastanow transition from a flat two-dimensional (2D) layer growth of homogenously strained In(Ga)As quantum wells to a self-organised formation of quantum dots occurs at a critical thickness that is itself temperature-dependent [9] but generally lies around 1.7 monolayers (ML) of indium (In) [10], however, it has been shown that a minimum critical indium concentration threshold must be reached in the deposition above, which indium segregation commences due to strain [11,12,13]. 

As the lattice parameter of an In_x_Ga_1−x_As alloy is given by *a*(*x*) = (0.56533 + 0.0405*x*) nm at room temperature [14], the relative misfit between the GaAs substrate and an In_x_Ga_1−x_As thin film at 300 K is [*a*_GaAs_ − *a*_InGaAs_ (*x*)]/*a*_GaAs_ = −0.0716*x*, hence the biaxial compressive stress increases with the total amount of In atoms deposited in the flux, i.e., linearly both with the increase of the average concentration, *x*, and with the total thickness, *d*, of the In(Ga)As layer. Capping In(Ga)As quantum dots by either binary GaAs or low concentration InGaAs alloys in order to flatten the growth surface for subsequent deposition of more quantum dots–as is necessary to advance from single-dot-layer pulsed lasers [15] to multiple-dot-layers continuous-wave lasers [16]–often takes a long time, and this article tries to explain why.

There are many reviews of In(Ga)As-based epitaxial quantum dot systems, the most comprehensive one probably being [7]; however, it is shown here that the models of homogeneous InAs quantum dots and the simplistic capping process used therein (figures 47, 62a and 64b in [7]) are incorrect. In fact, of the 14 most cited review articles on InGaAs quantum dots, the vast majority do not consider interdiffusion or segregation at all [7,17,18,19,20,21,22,23,24]. Overall, four reviews account for lateral adatom diffusion on surfaces [25,26,27,28], while those three that take into account possible segregation [26,27,29] refer to our previous studies, indicating that the role interdiffusion and segregation play during the complete epitaxial growth process is still not widely appreciated. We think any modelling without these effects will remain descriptive but fail to properly explain the underlying physics.

## 2. The Life Stages of an InGaAs Quantum Dot during Epitaxy

### 2.1. Prologue: Wetting Layer Formation (and Interdiffusion)

When InAs or InGaAs deposition commences on a flat, reconstructed GaAs(001) surface, then indium atoms tend to accumulate on the surface as their integration into the GaAs crystal lattice would require strain energy to be overcome. Thus, segregation drives the larger indium atoms to the free surface, while interdiffusion means In atoms could swap sites with underlying Ga atoms and be incorporated into the sublayer directly underneath. The result will be a very thin InGaAs wetting layer only a few monolayers thin.

### 2.2. Conception: Surface Corrugations (and Lateral Segregation due to Strain Build-Up)

When deposition continues and more In atoms accumulate on the surface, the surface starts to roughen, creating a topography of valleys and hillocks as shown in Figure 1, where troughs of the compressively strained valleys remain indium-poor while the material near the crests of the valleys can expand laterally to reduce strain–so the larger In atoms will accumulate here. These long-range and shallow surface corrugations are similar to the buckling that would happen if one tried to put a large carpet on the floor in a room a few centimeters too small for it. This surface waviness serves as a pre-cursor to the quantum dots that will develop later.

The three deposited InAs thin layers shown in Figure 2 have all been capped by GaAs, and only the top layer with the highest amount of indium shows clear quantum dots; the intermediate layer depicts some thickness fluctuations due to In/Ga exchange processes both during the deposition of InAs and the subsequent capping by GaAs. The lowest InAs layer appears widest due to vertical interdiffusion, which has effectively yielded a buried InGaAs thin film.

### 2.3. Birth: Formation of Small Quantum Dots (by Lateral Segregation due to Strain)

Figure 3 demonstrates that the onset of quantum dot formation occurs spontaneously when both a minimum InGaAs thickness (here: 3–4 nm) and a minimum indium concentration in the flux (here: *x* = 0.25) is reached, while for even only slightly lower fluxes (here: *x* = 0.24) the wafer surface remains mainly flat. For higher indium fluxes, the corresponding thickness at which the surface starts to roughen is correspondingly lower.

### 2.4. Growth: Expansion of Quantum Dots (by Lateral and Vertical Segregation)

Figure 4 shows an indium concentration map of one of those InGaAs quantum dots in Figure 3b that has been cross-sectioned close to its centre line. From this map one can clearly see
a.that the indium content near the centre of the quantum dot is more than twice as high as the deposited flux of 25% indium, due to vertical indium segregation,b.that the wetting layer is about 2–3 nm wide and has a lower indium content of only *x* ≈ 0.15 due to vertical Ga/In interdiffusion, andc.that the wetting layer has a further decreased indium content near the quantum dot (*x* ≈ 0.1) where the indium has been sucked up by the quantum dot. This lateral indium segregation correlates well with the darker rims observed around the larger quantum dots in Figure 3b.

**Figure 4 nanomaterials-12-03850-f004:**
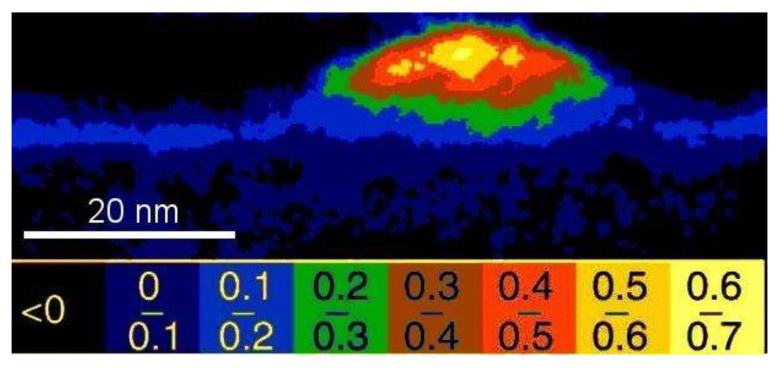
Indium concentration map from energy-filtered TEM (EFTEM) of an In_0.25_Ga_0.75_As quantum dot in cross-section, showing the indium depletion of the wetting layer and indium accumulation in the centre of the quantum dot by segregation. Reproduced with permission from [13] (greyscale) and [32] (colour).

Indium segregation from the highly stressed island edges towards the quantum dot centres can relieve misfit stress before misfit dislocations can nucleate there [33], which is important to suppress the generation of line defects that would be detrimental to the optical properties.

### 2.5. Demise: Trough Formation around Quantum Dots during GaAs Overgrowth

Figure 5 demonstrates that if In(Ga)As quantum dots are covered by thin GaAs cap layers, the initial growth proceeds mainly in the flat regions between the islands, sparing their highly strained edges where thus ridges several nanometres deep develop. These will only be filled in during later growth and are the main reason it takes much more GaAs deposition than anticipated based on the measured quantum dot height alone, typically at least 3–4 times their heights, before a surface sufficiently flat for subsequent layer growth is recovered.

### 2.6. Burial: Flattening of the Surface

Figure 6 compares stacks of multiple layers with quantum dots: in Figure 6a the quantum dots appear stacked on top of each other, at a slight angle, and the layers have a small remaining waviness to them, indicating the barrier layers in-between have not been thick enough to re-establish perfectly flat surfaces for subsequent layer growth. In Figure 6b, on the other hand, the InAs quantum dots were first embedded in InGaAs wells before much thicker GaAs barriers were deposited, thereby eliminating surface corrugations and corresponding lateral correlations between quantum dot positions in subsequent layers. 

## 3. Summary and Conclusions

The complex interplay between diffusion and segregation in InGaAs-based quantum dot systems has been described. Using energetic simulations of vertical segregation of In atoms, it has been shown [37] that if a critical indium flux is reached, segregation can trigger the spontaneous formation of islands, which relaxes some of the built-in strain energy and can therefore retard the alternative form of stress relief by nucleation of dislocations. The result will be an indium-depleted wetting layer and indium-enriched quantum dots, where some interdiffusion will usually prevent these from being pure binary InAs, even if that was originally deposited, which needs to be taken into account in simulations of the optical properties of such systems.

## Figures and Tables

**Figure 1 nanomaterials-12-03850-f001:**
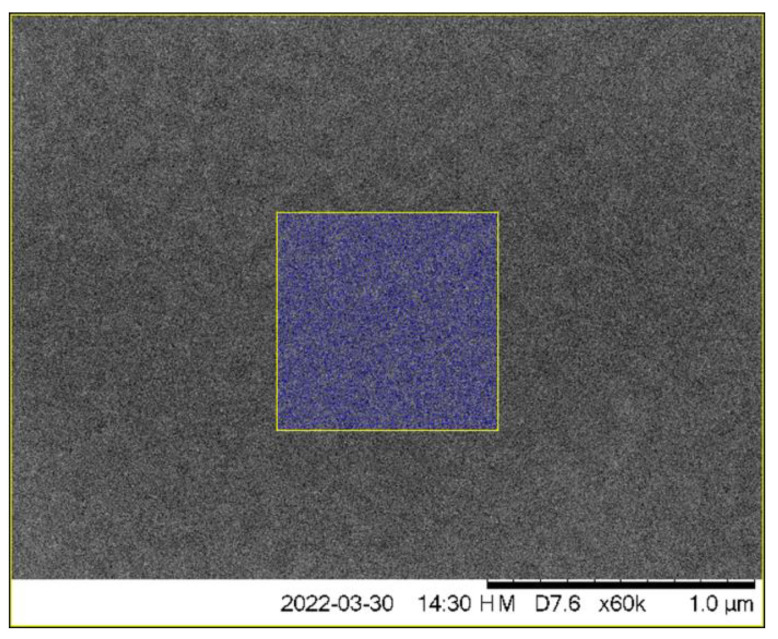
Back-scattered electron SEM image in plan-view or top-down geometry (black & white) with overlaid X-ray map (coloured inset; blue represents Ga and yellow In L-line intensities) of nominally 1.5 ML InAs on GaAs(001). Reproduced from [30].

**Figure 2 nanomaterials-12-03850-f002:**
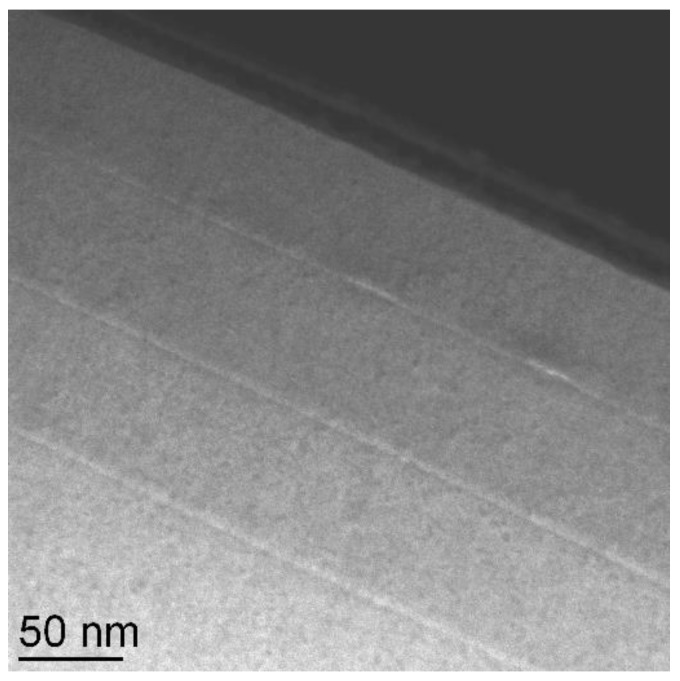
Annular dark-field (ADF) STEM image in cross-sectional view of three InAs thin films of nominal thicknesses of 1.6, 1.8, and 2.0 ML, i.e., near the Stranski-Krastanow transition. Quantum dots are only clearly visible in the top of the three InAs layers. Reproduced from [31].

**Figure 3 nanomaterials-12-03850-f003:**
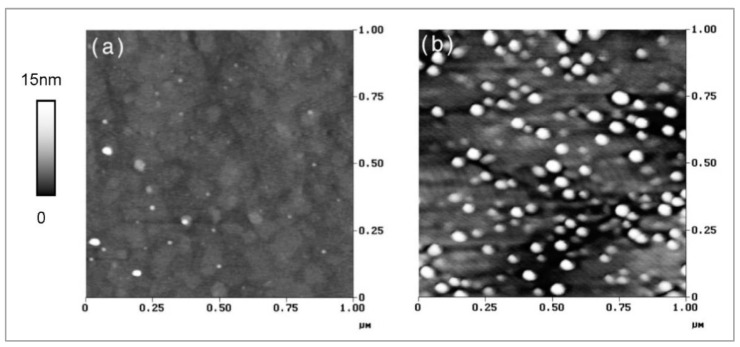
AFM topography maps of 3 nm (about 2.6 pure InAs equivalent ML) In_x_Ga_1−x_As layers with *x* = 0.24 (**a**) and *x* = 0.25 (**b**). Reproduced with permission from [11].

**Figure 5 nanomaterials-12-03850-f005:**
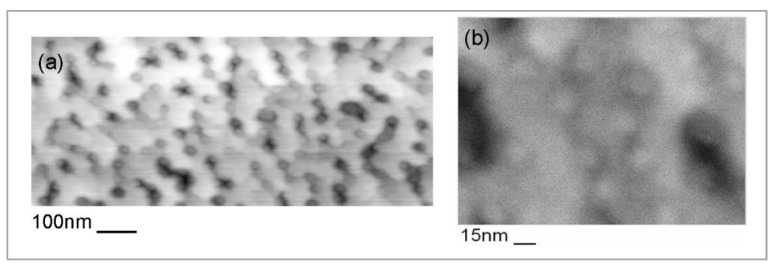
AFM topography (**a**) and ADF-STEM image in plan-view (**b**) of 8 nm GaAs overgrowth of In_0.25_Ga_0.75_As quantum dots, showing trench formation around them. (**a**) is a detail reproduced with permission from [34].

**Figure 6 nanomaterials-12-03850-f006:**
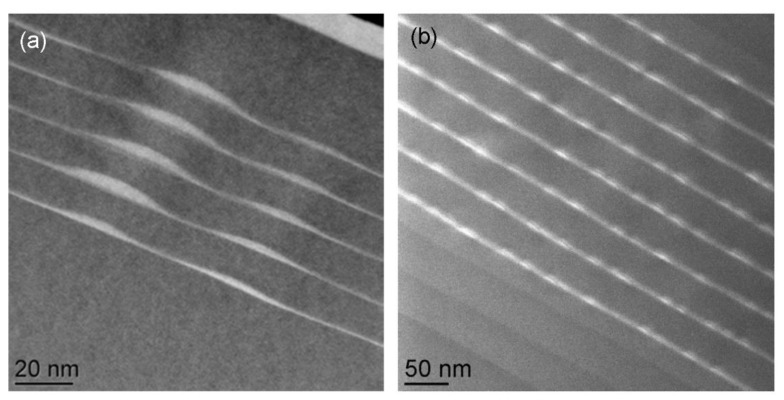
Cross-sectional ADF images of multi-layer stacks of (**a**) five repeats of 0.76 nm InP QDs with 12 nm AlInP barriers (reproduced from [35]) and (**b**) eight repeats of 0.8 nm (2.7 ML) InAs QDs with 5 nm InGaAs wells and 33 nm GaAs barriers from (reproduced from [36]).

## Data Availability

Not applicable.

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
