# Peer review of "Role of Interdiffusion and Segregation during the Life of Indium Gallium Arsenide Quantum Dots, from Cradle to Grave"

_nanomaterials, 2022, doi:10.3390/nano12213850_

Round 1

Reviewer 1 Report

The manuscript under consideration is indeed a nicely written work devoted to the state-of-the art characterization of InGaAs QD growth. Obviously, it is not a typical "Review paper", and it is more a tribute to a very notable person and decades of successful cooperation. While excessively brief and only providing general trends, the manuscript may be most useful for young scientists to spark research interest in the nanoworld. Thus, I would recommend to publish the work "as is".

The only concern I feel necessary to expess here refers to the title of the manuscript. In my opinion, it is rather misleading and should be reconsidered with account to the expected audience.

Thank you very much for your work!

Author Response

Reviewers 1 & 2 both had essentially the same comments, namely that the manuscript title was confusing and that the manuscript itself too short for a typical review. 
They are right, and this manuscript does not constitute a review in the classical sense. 
In response, the word 'review' has been deleted from both title and the abstract and replaced with a more descriptive statement concerning the topic of the manuscript. 
I have added a dozen of existing reviews to the bibliography and at the end of the introduction now use these to explain in more detail the aim of this study, which has also been added in brief to the first sentence in the abstract. 
I hope this addresses both issues and makes the manuscript more accessible to potential readers. All changes made have been marked by the 'track changes' option. 

Reviewer 2 Report

I have read with great attention the manuscript nanomaterials-1996408 “Review of the Life of Indium Gallium Arsenide Quantum Dots, from Cradle to Grave“ by Thomas Walther. The review devoted to the various aspects of the epitaxial InGaAs and InAs quantum dots.

As the type of reviewed manuscript declared as review, I found it too short and futile. The declared topic is popular in the scientific literature and count many hundred of manuscripts, but authors cite only 24. It is not clear what justified the choice of these particular articles and ignoring many similar ones. The motivation for writing this review also remains unclear. 

Nevertheless, I found this paper could be interesting for a broad community of readers’ Nanomaterial journal. The manuscript is clearly structures and well-written. Unfortunately, I could not recommend publish this review in the present form may be only after significant enhancement because title sounds intrigue.

Author Response

(The authors gave the same response as above.)

Reviewer 3 Report

1. There are a lot of language problems in the article, include tense, I hope the author can modify the language and typing problems in detail.

2. In the abstract, the authors should emphasize what results the characterizations indicate more specifically with crystalline parameters.

3. As authors mentioned gravel to grave it is better to add crystalline properties in comparison with the oxidation states so it will be easy for the reader to understand.

4. Quantum dots basically define the valence band theory and its particulates regarding lattice strain, dislocation density, and d-spacing parameters, depending upon these features the crystalline varies with respect to valence and covalent variations. Authors can specify clearly about the interaction and how the atomic properties variations happens in shell and core levels.

5. QDs also has a great tendency in fluorescence phenomena where the In and As deities interact with In by varying the Wykoff positions in first order and second order mediums, so it is better to add few particulates about this specific interactions.

6. A better understanding of AFM topography must be elaborated with respect to rigid body roughness and surface parameters of the QDs.

7. Author must carefully evaluate picture standards where all images must be same size and length with specifications of space bars below the numerical values. Eg.Fig.5.

8. over all the manuscript lacks standard to the title mentioned gravel to grave, which I sincerely suggest to add few more concepts regard of atomic properties of In towards Ga, As and their lateral interactions with pseudo-Voigt function and first order theory variations.

9. Authors must add recent studies and compare the outcome of the perspective with much deeper elaborations. Also, authors can site some recent publications,

Vilian, A. E., Veeramani, V., Chen, S. M., Madhu, R., Huh, Y. S., & Han, Y. K. (2015). Preparation of a reduced graphene oxide/poly-l-glutathione nanocomposite for electrochemical detection of 4-aminophenol in orange juice samples. Analytical Methods, 7(13), 5627-5634.

Veeramani, V., Chen, Y. H., Wang, H. C., Hung, T. F., Chang, W. S., Wei, D. H., ... & Liu, R. S. (2018). CdSe/ZnS QD@ CNT nanocomposite photocathode for improvement on charge overpotential in photoelectrochemical Li-O2 batteries. Chemical Engineering Journal, 349, 235-240.

Hwa, K. Y., Ganguly, A., Santhan, A., & Sharma, T. S. K. (2022). Synthesis of Water-Soluble Cadmium Selenide/Zinc Sulfide Quantum Dots on Functionalized Multiwalled Carbon Nanotubes for Efficient Covalent Synergism in Determining Environmental Hazardous Phenolic Compounds. ACS Sustainable Chemistry & Engineering, 10(3), 1298-1315.

Author Response

Reviewer 3  has come up with a number of detailed comments, many of which I think indicate some fundamental misunderstanding, so I will address them individually in the following:  

1. There are a lot of language problems in the article, include tense, I hope the author can modify the language and typing problems in detail.     
I have used the past tense only when discussing the content of previously published articles or when referring to the growth details of samples. All discussion is in the present tense, using future and conditional forms where appropriate.  Two typos have been corrected.
2. In the abstract, the authors should emphasize what results the characterizations indicate more specifically with crystalline parameters.
I do not understand what the reviewer wants to say: the lattice parameters used are stated on top of page 2.    
3. As authors mentioned gravel to grave it is better to add crystalline properties in comparison with the oxidation states so it will be easy for the reader to understand. 
A 'cradle' means the place where a newborn baby is kept, here it refers to the origin of quantum dot formation, and has nothing to do with 'gravel'. As epitaxy is typically performed under ultra-high vacuum conditions, there are no problems with oxidation.
4. Quantum dots basically define the valence band theory and its particulates regarding lattice strain, dislocation density, and d-spacing parameters, depending upon these features the crystalline varies with respect to valence and covalent variations. Authors can specify clearly about the interaction and how the atomic properties variations happens in shell and core levels.
The electronic structure of quantum dots can certainly be described by band theory but this report does not deal with such.
5. QDs also has a great tendency in fluorescence phenomena where the In and As deities interact with In by varying the Wykoff positions in first order and second order mediums, so it is better to add few particulates about this specific interactions.
I am not sure what the reviewer means by 'particulates'; here, we deal not with any colloidal particles but with coherently strained epitaxial islands, as long as no dislocations are generated to relax stress.
6. A better understanding of AFM topography must be elaborated with respect to rigid body roughness and surface parameters of the QDs.
Our AFM images simply show height variations on the surfaces examined - this is standard topography contrast.  
7. Author must carefully evaluate picture standards where all images must be same size and length with specifications of space bars below the numerical values. Eg.Fig.5.
Images that show phenomena at different length scales cannot be shown at the same size, and all the micrographs shown here do have scale bars from which the reader can work out sizes in real space.
8. over all the manuscript lacks standard to the title mentioned gravel to grave, which I sincerely suggest to add few more concepts regard of atomic properties of In towards Ga, As and their lateral interactions with pseudo-Voigt function and first order theory variations.
For band structure theory references [7] and [29] may be perused but this is not the topic of this article, cf. point 4.
9. Authors must add recent studies and compare the outcome of the perspective with much deeper elaborations. Also, authors can site some recent publications,
The below references are all to applications of colloidal quantum dots made up of nano-crystals from II/VI compound semiconductors produced by wet chemistry in an organic chemistry lab. This has nothing to do with the topic of this article, which is on fundamental growth aspects of epitaxial quantum dots of III/V compound semiconductors created by physical deposition methods under ultra-high vacuum conditions. The below studies therefore do not fit; however, I have instead included references to a dozen more review articles to provide better context. 
Vilian, A. E., Veeramani, V., Chen, S. M., Madhu, R., Huh, Y. S., & Han, Y. K. (2015). Preparation of a reduced graphene oxide/poly-l-glutathione nanocomposite for electrochemical detection of 4-aminophenol in orange juice samples. Analytical Methods, 7(13), 5627-5634.
Veeramani, V., Chen, Y. H., Wang, H. C., Hung, T. F., Chang, W. S., Wei, D. H., ... & Liu, R. S. (2018). CdSe/ZnS QD@ CNT nanocomposite photocathode for improvement on charge overpotential in photoelectrochemical Li-O2 batteries. Chemical Engineering Journal, 349, 235-240
Hwa, K. Y., Ganguly, A., Santhan, A., & Sharma, T. S. K. (2022). Synthesis of Water-Soluble Cadmium Selenide/Zinc Sulfide Quantum Dots on Functionalized Multiwalled Carbon Nanotubes for Efficient Covalent Synergism in Determining Environmental Hazardous Phenolic Compounds. ACS Sustainable Chemistry & Engineering, 10(3), 1298-1315.

Round 2

Reviewer 2 Report

I have read a revised manuscript nanomaterials-1996408 "Role of Interdiffusion and Segregation during the Life of Indium Gallium Arsenide Quantum Dots, from Cradle to Grave" by Thomas Walther. Indeed, the author replaced "review" word in the Title. However, manuscript type is still mentioned as a review (see attached file). This fact is very confusing. I stay in my previous opinion that at the present form manuscript is too short as a review. The previous remark: "The motivation for writing this review also remains unclear" stays unanswered even after revision. 

This paper could be interesting for a broad community of readers’ Nanomaterial journal after significant expansion. 

Author Response

I have contacted the assistant editor to ensure the manuscript type is consistently changed from review to article.

The motivation for our work really is that, while we have been working on investigating atomic diffusion and segregation in this system for about twenty years, we could close some gaps in our understanding only recently:

Figure 1 is a plan-view SEM image & map for which we needed to operate a large SDD X-ray detector in a medium-voltage SEM for more than an hour to prove indium fluctuations altready exist in a just-formed wetting layer.

Figure 5b is a Z-contrast STEM image of a very thin region with several InGaAs quantum dots that had been capped by nominally 8nm GaAs, which confirms the existence of trenches observed also in AFM (Fig. 5a) around the islands' bases that remain essentially uncapped.

Both experiments now fill the previously existing gaps in experimental evidence for the interplay of segregation and strain at the onset of formation of the wetting layer as well as at the final stage of quantum dot capping, which motivated this article. 

Reviewer 3 Report

I accept the manuscript in the current form

Author Response

This reviewer has accepted all changes made to our manuscript so there are no outstanding tasks - many thanks!